# Durability of Humoral and Cellular Immunity after an Extended Primary Series with Heterologous Inactivated SARS-CoV-2 Prime-Boost and ChAdOx1 nCoV-19 in Dialysis Patients (ICON3)

**DOI:** 10.3390/vaccines10071064

**Published:** 2022-07-01

**Authors:** Sarinya Boongird, Chavachol Setthaudom, Rungthiwa Kitpermkiat, Somsak Prasongtanakij, Supanart Srisala, Piyatida Chuengsaman, Arkom Nongnuch, Montira Assanatham, Sasisopin Kiertiburanakul, Kumthorn Malathum, Angsana Phuphuakrat, Jackrapong Bruminhent

**Affiliations:** 1Division of Nephrology, Department of Medicine, Faculty of Medicine Ramathibodi Hospital, Mahidol University, Bangkok 10400, Thailand; sarinya.bon@mahidol.ac.th (S.B.); rungthiwa.k@outlook.com (R.K.); oatega@yahoo.com (A.N.); montira.asa@mahidol.ac.th (M.A.); 2Immunology Laboratory, Department of Pathology, Faculty of Medicine Ramathibodi Hospital, Mahidol University, Bangkok 10400, Thailand; chavachol@hotmail.com; 3Office of Research, Academic Affairs and Innovation, Faculty of Medicine Ramathibodi Hospital, Mahidol University, Bangkok 10400, Thailand; somsak.pra@mahidol.edu (S.P.); supanart.sri@mahidol.ac.th (S.S.); 4Banphaeo-Charoenkrung Peritoneal Dialysis Center, Banphaeo Dialysis Group, Banphaeo Hospital, Bangkok 10120, Thailand; pjeungsmarn@yahoo.com; 5Division of Infectious Diseases, Department of Medicine, Faculty of Medicine Ramathibodi Hospital, Mahidol University, Bangkok 10400, Thailand; sasisopin.kie@mahidol.ac.th (S.K.); mkumthorn@yahoo.com (K.M.); angsana.phu@mahidol.ac.th (A.P.)

**Keywords:** COVID-19 vaccine, dialysis, immune response, inactivated SARS-CoV-2 vaccine, neutralizing antibody, SARS-CoV-2, viral vector vaccine, waning immunity

## Abstract

The durability of a three-dose extended primary series of COVID-9 vaccine in dialysis patients remains unknown. Here, we assessed dynamic changes in SARS-CoV-2-specific humoral and cell-mediated immunity at baseline, 3 months, and 6 months after the extended primary series in 29 hemodialyzed (HD), 28 peritoneal dialyzed (PD) patients, and 14 healthy controls. Participants received two doses of inactivated SARS-CoV-2 vaccine followed by a dose of ChAdOx1 nCoV-19 vaccine. At 6 months, median anti-RBD IgG titers (IQR) significantly declined from baseline in the HD (1741 (1136–3083) BAU/mL vs. 373 (188–607) BAU/mL) and PD (1093 (617–1911) BAU/mL vs. 180 (126–320) BAU/mL) groups, as did the mean percent inhibition of neutralizing antibodies (HD: 96% vs. 81%; PD: 95% vs. 73%) (all *p* < 0.01). Age and post-vaccination serological response intensity were predictors of early humoral seroprotection loss. In contrast, cell-mediated immunity remained unchanged. In conclusion, humoral immunity declined substantially in dialysis patients, while cell-mediated immunity remained stable 6 months after the extended heterologous primary series of two inactivated SARS-CoV-2/ChAdOx1 nCoV-19 vaccine. A booster dose could be considered in dialysis patients 3 months after this unique regimen, particularly in the elderly or those with a modest initial humoral response.

## 1. Introduction

The coronavirus disease 2019 (COVID-19) pandemic poses challenges for dialysis patients globally. Dialysis patients are at increased risk of infection and mortality from severe acute respiratory syndrome coronavirus 2 (SARS-CoV-2), owing to immunological dysfunction and comorbidities [1,2]. Despite the impaired vaccine response in dialysis patients, COVID-19 immunization reduces the COVID-19 incidence and hospitalization rate in this population [3,4]. In comparison with other vaccine platforms, the inactivated SARS-CoV-2 vaccine appears to induce weaker humoral immunity (HMI) in immunocompromised persons [5], including those on dialysis [6,7,8]. Hence, the World Health Organization (WHO) recommends the members of vulnerable groups receive an additional COVID-19 vaccine dose after their primary vaccination series [9]. Third-dose vaccine, either homologous or heterologous, can significantly enhance HMI and cell-mediated immunity (CMI) in both the general population and dialysis patients compared with a standard two-dose primary series [10,11,12,13].

Nevertheless, a substantial decline in immunity and vaccine effectiveness has been documented in the general population as early as 3 months after a homologous three-dose mRNA vaccine [14]. Similarly, diminished immunity has been reported in dialysis patients a few months following a two-dose primary series of COVID-19 vaccine across vaccination platforms, with most data coming from mRNA vaccines [15,16,17]. Limited data exists on the longevity of immune responses after a third COVID-19 vaccine dose in the dialysis population, particularly for heterologous three-dose regimens. Such information is essential for determining the optimal time to provide a fourth dose for restoring protective immunity. Here, we evaluated the HMI and CMI durability following an extended heterologous primary vaccination series with two doses of inactivated SARS-CoV-2 and a third dose of ChAdOx1 nCoV-19 in dialysis patients and healthy volunteers. Predictors of early humoral seroprotection loss in these vulnerable individuals were also investigated.

## 2. Materials and Methods

### 2.1. Study Design

We conducted a prospective cohort study from July 2021–January 2022 at the Faculty of Medicine Ramathibodi Hospital, Mahidol University in Bangkok, Thailand. Participants from the ICON1 [6] and ICON2 [10] studies were screened and enrolled. They had received a primary series of two-dose CoronaVac vaccine (3 μg inactivated SARS-CoV-2 whole virus in 0.5 mL; Sinovac Biotech Ltd., Beijing, China) (ICON1) followed by one dose of ChAdOx1 nCoV-19 vaccine (AZD1222, ChAdOx1-S at a minimum concentration of 2.5 × 10^8^ infectious units per 0.5 mL), with doses administered 4 weeks apart (ICON2), referred to throughout as the extended heterologous primary series.

Eligible participants were dialysis patients aged 18–59 years old who had been stable for at least 1 month on their hemodialysis (HD) or peritoneal dialysis (PD) prescriptions. Healthy volunteers with normal kidney function were recruited from our community, and they were similar in age to the dialysis group members. Participants were excluded if they had active illnesses, were diagnosed with COVID-19, or had experienced any respiratory tract symptoms 3 days before enrollment. SARS-CoV-2 reverse transcription-polymerase chain reaction (RT-PCR) testing on nasopharyngeal and oropharyngeal swabs was not performed before study inclusion. Patients who received COVID-19 immunizations outside of the study or were diagnosed with COVID-19 infection during the study were excluded from the analysis.

We examined the HMI and CMI dynamics in patients undergoing HD and PD at 2 weeks (M0), 3–4 months (M3), and 6 months (M6) after receiving the dose of ChAdOx1 nCoV-19 vaccine (Figure 1). In healthy controls, the HMI and CMI were assessed only at M0 and M3. HMI was assessed with the SARS-CoV-2 immunoglobulin G (IgG) assay and the SARS-CoV-2 surrogate viral neutralization test (sVNT), which, respectively, determine the IgG antibody titer and functional neutralizing antibody activity against the S1 receptor-binding domain (RBD) of the SARS-CoV-2 spike protein. CMI was determined using an enzyme-linked immunospot assay for interferon (IFN)-γ.

### 2.2. SARS-CoV-2-Specific Humoral Immunity

The Abbott SARS-CoV-2 IgG II Quantification test (Abbott Diagnostics, Lake Bluff, IL, USA) was used to determine the concentration of anti-RBD IgG against SARS-CoV-2. Plasma samples were analyzed in accordance with the manufacturer’s recommendations using the Abbott Alinity system. Anti-RBD IgG concentrations are presented in binding antibody units (BAU)/mL. Seroconversion was defined as an anti-RBD IgG titer of ≥7.1 BAU/mL [18]. The diagnostic sensitivity and specificity of this cut-off were 91.6% and 99.4%, respectively [18,19]. This study used an anti-RBD IgG cut-off value of >506 BAU/mL for seroprotection because it corresponded to 80% protection against symptomatic COVID-19 infection when the SARS-CoV-2 Alpha variant (B.1.1.7) was dominant [20]. The dynamic changes in HMI over time were estimated by dividing the anti-RBD IgG titer differences (BAU/mL) between M0 and M3 or M0 and M6 by the anti-RBD IgG titer at M0, and then expressing the results as percent changes.

A SARS-CoV-2 NeutraLISA surrogate neutralization assay (Euroimmun, Lübeck, Germany) was used to evaluate the function of anti-SARS-CoV-2 S1/RBD antibodies. This test detects plasma neutralizing antibodies that compete with the RBD for binding to the angiotensin-converting enzyme 2 receptor protein. The RBD sequence was obtained from the WH-Human 1 coronavirus. The neutralizing antibody level was quantified and expressed as percent inhibition. Participants with a percent inhibition of >35% were classified as neutralizing antibody positive [21]. This value has a specificity of 99.7% and a sensitivity of 95.9% [22].

### 2.3. SARS-CoV-2-Specific Cell-Mediated Immunity

To quantify SARS-CoV-2-specific vaccine-induced CMI, we used the SARS-CoV-2 ELISpot test; the details of this technique have been previously described [23]. Anti-CD3 antibody was used as a positive control, and a SARS-CoV-2 S1 scanning peptide pool (Mabtech, Stockholm, Sweden), derived from the WH-Human 1 coronavirus, was used as a stimulant. For the S1 scanning peptide pool assay, the results are presented as IFN-producing spot-forming units (SFU) per 10^6^ peripheral blood mononuclear cells (PBMCs). Responders were those who had at least 6 SFU/10^6^ PBMCs [24,25].

### 2.4. Statistical Analyses

For baseline characteristics, categorical data are expressed as absolute numbers (%), while continuous variables are reported as mean (standard deviation (SD)) or median (interquartile range (IQR)), depending on the data distribution. The independent sample *t*-test or Mann–Whitney U test was used to evaluate the difference between groups for continuous variables, and the chi-square or Fisher’s exact test was used for categorical variables. To compare continuous variables among more than two groups, an analysis of variance was used. We used a stepwise logistic regression technique to identify predictors of anti-RBD IgG seroprotection status 3 months after vaccination completion. In a multivariate logistic regression model, any variables with a *p*-value of <0.1 from the univariate analysis and serum albumin were included. We used a multilevel mixed-effects linear regression model to compare the dynamic changes in HMI and CMI at various timepoints within and across groups. Comparisons of proportions of patients achieving specific immune thresholds at different timepoints within and across groups were analyzed with a mixed logistic model. A *p*-value of < 0.05 was considered statistically significant. Data analysis was performed using Stata statistical software, version 17 (StataCorp, LLC; College Station, TX, USA). The prevalence of anti-RBD IgG seroconversion, anti-RBD IgG seroprotection, percent neutralization antibody positivity, and S1-specific T-cell responders are presented as bar graphs created with GraphPad Prism 9.0 (GraphPad Software, Inc.; San Diego, CA, USA).

## 3. Results

### 3.1. Participant Clinical Characteristics

There were 75 eligible participants from the ICON2 study, all of whom received a primary two-dose series of inactivated SARS-CoV-2 plus a third dose of the ChAdOx1 nCoV-19 vaccine. This study (ICON3) had 71 participants: 29 patients undergoing HD, 28 patients undergoing PD, and 14 healthy control participants (Figure 1). All study participants underwent HMI and CMI assessment at 3–4 months (M3) after receiving the ChAdOx1 nCoV-19 vaccine, with 50 dialysis patients (88% of study participants) undergoing another HMI and CMI assessment at 6 months (M6). The mean time intervals from M0 to M3 and from M0 to M6 were similar in the HD and PD groups (Table 1). Participants in the control group underwent immunogenicity evaluation (M3) later compared with those in the dialysis group, with a mean (SD) time interval from M0–M3 of 4.2 (0.3) months vs. 3.4 (0.2) months, *p* < 0.01, respectively. Baseline demographic and laboratory parameters of the study participants are shown in Table 1. Patient age, sex, comorbidities, body mass index, dialysis vintage, and most basic laboratory parameters were comparable between the two dialysis groups. However, patients undergoing HD had higher mean (SD) serum albumin (40.1 (4.3) g/L vs. 33.1 (4.1) g/L, *p* < 0.01) but lower median (IQR) serum ferritin (304 (119–441) ng/mL vs. 367 (156–751) ng/mL, *p* = 0.04) compared with patients undergoing PD. Control participants had a mean (SD) age of 43.6 (8) years, and 36% were men. Patients undergoing HD or PD were more likely to be male and had a higher prevalence of diabetes mellitus compared with control participants. Age was comparable among all groups. Only one patient in the PD group was taking 5 mg of prednisolone per day. None of the participants in the control group were immunocompromised or taking immunosuppressants.

### 3.2. SARS-CoV-2-Specific MHI

#### 3.2.1. Anti-RBD IgG

The dynamic changes in HMI at 3–6 months after completing the extended heterologous primary series are illustrated in Figure 2. At 2 weeks after the ChAdOx1 nCoV-19 vaccine (M0), patients in the HD and control groups achieved comparable titers of anti-RBD IgG (median (IQR), HD: 1741 (1136–3083) BAU/mL vs. control: 2269 (1607–2830) BAU/mL) (Figure 2a). However, patients in the PD group had a significantly lower median anti-RBD IgG titer (1093 (617–1911) BAU/mL) compared with the other two groups (*p* < 0.01).

At M3, the median anti-RBD IgG titer declined significantly from the M0 titer in all groups (*p* < 0.01). The median (IQR) anti-RBD IgG titer at M3 in the HD group was significantly higher compared with the PD group (632 (337–1526) BAU/mL vs. 368 (217–5540) BAU/mL, *p* = 0.02). In the control group, the median (IQR) anti-RBD IgG titer at M3 was 482 (196–658) BAU/mL, which did not significantly differ from either the HD or PD groups. The mean (SD) percent reduction in anti-RBD IgG titer from M0 to M3 was greatest in the control group at −80.2% (11%). Compared to the control group, the mean (SD) percent reduction in anti-RBD IgG titer from M0 to M3 was significantly lower in the HD group (−59.5% (27.6%), *p* = 0.01), whereas it was marginally lower in the PD group (−69.1% (17.1%), *p* = 0.07). The mean (SD) percent reduction of anti-RBD IgG from M0 to M3 in patients undergoing HD was comparable to that in patients undergoing PD (*p* = 0.13) (Appendix A). If the M0–M3 time interval differences between groups are taken into account, anti-RBD IgG declined at comparable rates across all groups (*p* = 0.19). The mean (SD) anti-RBD IgG decay rate over time, calculated by dividing the percent reduction of anti-RBD IgG from M0 to M3 by the time interval from M0 to M3, was −0.6% (0.28%)/day in the HD, −0.7% (0.2%)/day in the PD, and −0.7% (0.2%)/day in the control group. Anti-RBD IgG seroconversion rates remained constant at 100% in all groups (Figure 3a). The proportions of patients with an anti-RBD IgG titer higher than the seroprotective threshold against the SARS-CoV-2 Alpha variant (>506 BAU/mL) at various timepoints are shown in Figure 3b. Patients in the HD group were more likely to be seroprotected than were patients in the PD group (59% vs. 32%, *p* = 0.04) at M3.

At M6, the median (IQR) anti-RBD IgG titers continued to decline in both the HD (M0: 1741 (1136–3083) BAU/mL vs. M6: 373 (188–607) BAU/mL, *p* < 0.01) and PD groups (M0: 1093 (617–1911) BAU/mL vs. M6: 180 (126–320) BAU/mL, *p* < 0.01). There was no difference in the median anti-RBD IgG titer or anti-RBD IgG percent reduction between the HD and PD groups. Anti-RBD IgG seroconversion rates remained stable in all groups, with no difference in anti-RBD IgG seroconversion rates observed between the HD and PD groups at any timepoint (Figure 3a). The percentage of participants with anti-RBD IgG titers above the seroprotective threshold decreased over time in both the HD and PD groups (*p* < 0.01 for all timepoints) (Figure 3b). Notably, 9 (36%) of the HD patients and no PD patients had anti-RBD IgG titers over the seroprotective threshold at M6 (*p* < 0.01).

#### 3.2.2. Neutralizing Antibody

Neutralizing antibody activity, as measured by the percent (%) inhibition in a neutralization (NT) test, decreased over time (from M0 to M6) in all groups (*p* < 0.01) (Figure 2b). At all timepoints, patients undergoing HD had mean (SD) % inhibition comparable to that of patients undergoing PD. Likewise, control participants had mean % inhibitions at M0 and M3 similar to those of patients undergoing HD or PD (Figure 2b). All patients were neutralizing antibody seropositive at baseline. The neutralizing antibody seropositive rates were maintained at M3 and M6 in the HD (93% and 88%, respectively; *p* = 0.72) and PD (82% and 80%, respectively; *p* = 0.79) groups. There was no difference in median percent reduction of neutralizing antibodies from M0 to M3 or M0 to M6 between the HD and PD groups (Appendix A). The median percent reduction of neutralizing antibody and neutralizing antibody positivity rate from M0 to M3 were comparable between the dialysis and control groups (Appendix A and Figure 3c).

#### 3.2.3. Predictors of Early Loss of Anti-RBD IgG Seroprotection Status among Dialysis Patients at M3

At M3, 31 (57%) dialysis group patients had lost anti-RBD IgG seroprotection status. Predictors of early anti-RBD IgG seroprotection loss at M3 are presented in Table 2. In a multivariate logistic regression model, age and anti-RBD IgG tertile at M0 were independent predictors of early loss of anti-RBD IgG seroprotection status after adjusting for dialysis vintage, dialysis modality, and serum albumin level. The chances of losing anti-RBD IgG seroprotection status at M3 increased by 10% for each additional year of age (odds ratio (OR): 1.10; 95% confidence intervals (CI): 1.01–1.19; *p* = 0.03). Additionally, patients in the highest anti-RBD IgG tertile at M0 were 94% less likely to lose anti-RBD IgG seroprotection status compared with those in the first tertile (OR: 0.06; 95% CI: 0.01–0.38; *p* < 0.01).

### 3.3. SARS-CoV-2-Specific CMI

The dynamic SARS-CoV-2-specific CMI changes following the extended heterologous primary series are presented in Figure 2c. At M0, all groups had a similar median (IQR) number of SFUs for S1-specific T-cell responses. At M6, the median (IQR) number of SFUs for S1-specific T-cell responses remained unchanged and comparable in the HD (M0: 188 (32–480) vs. M6: 160 (72–396) SFU/10^6^ PBMCs) and PD groups (M0: 242 (74–470) vs. M6: 240 (108–480) SFU/10^6^ PBMCs) (Figure 2c). Similarly, the proportions of S1-specific T-cell responders were comparable between the HD and PD groups at M0 (93% vs. 96%, *p* = 0.99) and M6 (88% vs. 96%, *p* = 0.61) and remained relatively stable over time (Figure 3d). The median (IQR) SFUs for S1-specific T-cell responses increased slightly in the control group from M0 to M3 (105 (20–206) vs. 322 (228–480), *p* = 0.15), but did not differ significantly from the predicted responses in the dialysis group.

## 4. Discussion

This study evaluated the longevity of SARS-CoV-2-specific HMI and CMI up to 6 months following the extended heterologous primary COVID-19 vaccination series in patients undergoing dialysis and healthy volunteers. Although SARS-CoV-2-specific HMI remained detectable in most dialysis patients at 6 months after this unique regimen, both dialysis patients and healthy volunteers demonstrated a significant decline in HMI intensity of >75%, with similar decline rates for dialysis patients and control participants. Anti-RBD IgG titers and seroconversion rates in HD patients were higher than those in PD patients at M3 and M6. However, after adjusting for other variables, the impact of dialysis modality disappeared. Higher age and lower intensity of serological response during the initial post-vaccination period were independent predictors of early humoral seroprotection status loss. Conversely, CMI in dialysis patients remained stable at 6 months after this heterologous regimen.

Increasingly more data are available regarding the longevity of SARS-CoV-2-specific HMI in dialysis patients following a primary vaccination series or a booster dose. Most studies indicate that vaccine-induced HMI decreases significantly by 6 months following the primary series, regardless of vaccine type, in both dialysis patients and the general population [15,26,27,28,29,30]. HMI waning begins as early as 3 weeks after vaccination and gradually decreases over time [15,16,29,31]. Anand and colleagues monitored anti-RBD IgG levels monthly in 2563 dialysis patients who received mRNA-1273 or Ad26.COV2.S vaccine. They discovered that the proportion of patients with undetectable anti-RBD IgG increased over time, from 8.5% at 2–3 months to 20.2% 5–6 months later, with the lowest seronegative rate in patients receiving mRNA-1273 and the highest in patients receiving Ad26.COV2.S [15]. While the inactivated SARS-CoV-2 vaccine is widely used worldwide, it has been shown to elicit weaker HMI in immunocompromised individuals compared with other vaccine platforms [5,10]. Although comparing immunogenicity across studies using a variety of assays, cutoff values, and patient populations is challenging, we demonstrated with the same serological assay that our extended heterologous primary series could generate at least comparable anti-RBD IgG titer and seroconversion rate at 6 months post-vaccination to those induced by the standard two-dose mRNA vaccination [27]. The effect of dialysis modality on SARS-CoV-2-vaccine-induced antibody levels remains uncertain. In several studies, PD patients exhibited higher vaccine-induced antibody levels than HD patients [31,32]. However, in our cohort, we observed lower antibody titers in PD patients compared with HD patients, which could be explained by a lower serum albumin level, a well-known risk factor for reduced HMI for COVID-19 and other vaccines in the PD group [4,28,33]. Notably, the mean anti-RBD IgG decay rate was comparable in HD and PD patients, as well as in healthy volunteers, ranging from 0.6–0.7% reduction per day, which is consistent with previous reports following a two-dose regimen of BNT162b2 [26,27].

Although the seroconversion rate remained excellent in both HD and PD patients at M6, the magnitude of anti-RBD IgG titer significantly declined at 3 months after vaccination completion. Thus, we examined predictors of early seroprotection loss, using a cutoff established by Feng et al. [20], to identify patients who would benefit from an early booster. Age and peak HMI intensity during the post-vaccination period were independent predictors of early seroprotection loss. These predictors support findings from other reports [15,29] and may explain why the PD patients in this study, who had a substantially lower anti-RBD IgG titer following M0, lost seroprotection at a greater rate compared with HD patients. Dialysis modality and vintage were found to be significant in univariate analysis until the data are adjusted for anti-RBD-IgG titer. At M6, less than one-third of dialysis patients remained seroprotected. Our findings corroborate and strongly support the CDC recommendation [34] that a fourth dose of COVID-19 vaccine should be considered at least 3 months following the third dose in moderately or severely immunocompromised individuals.

Immune protection against COVID-19 is inversely proportional to the neutralizing antibody levels [35]. As with the anti-RBD seroconversion rate, the neutralizing antibody positivity rate declined over time. However, >80% of dialysis patients remained seropositive for neutralizing antibodies at 6 months after completing the extended heterologous primary series. This value is greater than the rate observed with prime-boost mRNA vaccination, which varied between 45% and 77% at 3 months post-vaccination [26,31]. Of note, the neutralizing antibody seropositive rate at M3 in our healthy volunteers (100%) was similar to those reported by other studies using prime-boost mRNA vaccination [26].

Recent studies demonstrated that the induced CMI following a primary series of SARS-CoV-2 vaccination declines over time [27,36]. In one large cohort of HD patients, the mean (95% CI) percent reduction in induced CMI, measured by IFN-γ secretion in response to antigen 2, from weeks 8/9 to 24 was −22.1% (−12.8% to −30.4%) and −45.9% (−38.5% to −52.4%) for dialysis patients who received two doses of BNT162b and mRNA-1273, respectively [27]. In contrast, we found that at 6 months after vaccination with a heterologous inactivated SARS-CoV-2 virus/ChAdOx1 nCoV-19 regimen, CMI in HD and PD patients remained robust. Dialysis patients in our study were relatively young, with a low dialysis vintage time, compared with other trials. Furthermore, delivering the whole virus to the immune system, either naturally by infection or possibly via an inactivated SARS-CoV-2 vaccine, might affect the duration or specifics of the induced CMI [4,37,38]. Some studies found that adenovirus-based platforms induce a greater spike-specific T-cell response compared with mRNA-based platforms [39,40]. Hence, among populations with impaired T-cell function, such as dialysis patients, our heterologous inactivated SARS-CoV-2 virus/ChAdOx1 nCoV-19 vaccination regimen may seem appealing. Nevertheless, the roles and implications of the induced CMI in preventing SARS-CoV-2 infection or reinfection remain incompletely explored [41].

This is one of the first studies evaluating the long-term durability of HMI and CMI induced by extended heterologous primary series in patients undergoing HD or PD. Additionally, immune responses in these patients were compared with those of healthy volunteers at 3 months following complete vaccination. For HMI, we assessed not only anti-RBD IgG but also neutralizing antibodies, which more accurately reflect the real function of the anti-SARS-CoV-2 S1/RBD antibody. Moreover, predictors of early seroconversion loss, which may help identify vulnerable individuals who could benefit from receiving an early booster, were determined. Our study has several limitations. Due to the small number of patients, its findings and conclusions should be considered preliminary. Further studies with larger sample sizes are required to determine definitive conclusions. Because we evaluated the immune response in healthy controls for only 3 months, we were unable to compare the durability of vaccine-induced immunity between dialysis patients and healthy volunteers over a longer duration. Moreover, the serological assays used were developed primarily for the original wild-type virus, and the seroprotection thresholds implemented here were derived from the cutoff value against the SARS-CoV-2 Alpha variant. Therefore, extrapolations regarding immunogenicity against newly evolving SARS-CoV-2 strains should be interpreted cautiously. Additionally, we did not check for natural infection, so there is a possibility that natural infection could have occurred and boosted participant immunity.

In conclusion, our study demonstrated a substantial decline in vaccine-induced HMI but a relative preservation of vaccine-induced CMI in dialysis patients at 6 months after the extended heterologous primary series (two doses of inactivated SARS-CoV-2, followed by a third dose with ChAdOx1 nCoV-19). In settings where a heterologous inactivated SARS-CoV-2/ChAdOx1 nCoV-19 vaccine regimen is implemented, we propose that a booster dose be considered at 3 months afterward to enhance the immunological response, especially in elderly persons or those with a poor antibody response during the early post-vaccination period. Further studies with larger populations are encouraged to better generalize our results and evaluate the clinical effectiveness of this vaccine regimen.

## Figures and Tables

**Figure 1 vaccines-10-01064-f001:**
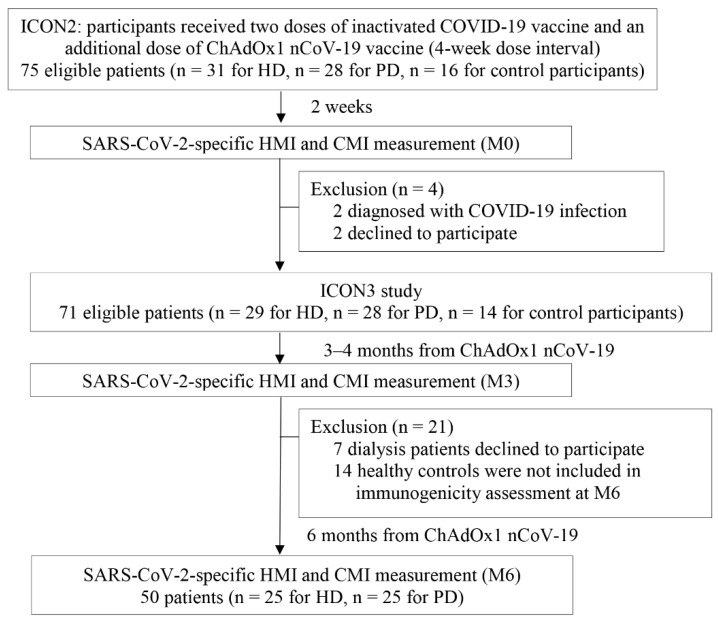
Study flowchart. CMI, cell-mediated immunity; COVID-19, coronavirus disease 2019; HD, hemodialyzed patients; HMI, humoral immunity; IgG, immunoglobulin G; n, number; SARS-CoV-2, severe acute respiratory syndrome coronavirus-2.

**Figure 2 vaccines-10-01064-f002:**
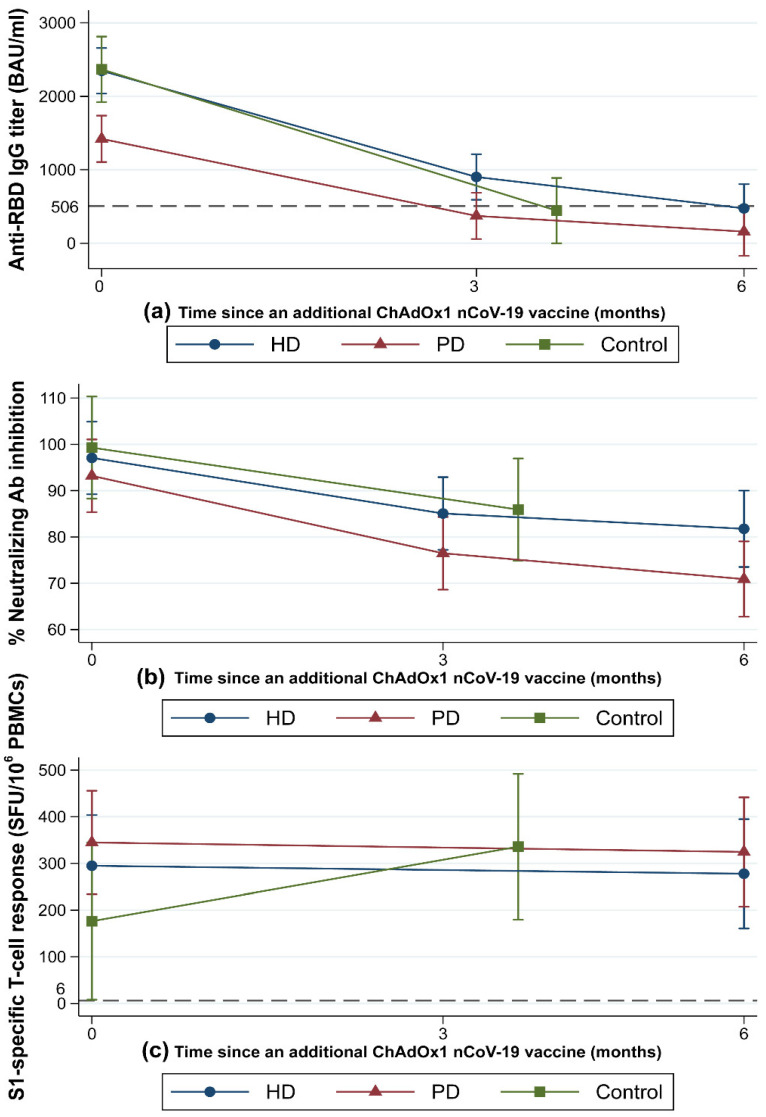
Dynamic changes in immunity following COVID-19 vaccination with an extended primary series. Patients undergoing HD (blue solid line) or PD (red solid line) and control participants (green solid line) were vaccinated with a heterologous inactivated SARS-CoV-2 prime-boost and then received a dose of ChAdOx1 nCoV-19. Dynamic changes in immunity were monitored from baseline (n = 57 for dialysis patients and 14 for controls) to 3 months (n = 57 for dialysis patients and 14 for controls) and 6 months (n = 50 for dialysis patients) after the third dose. (**a**) Geometric mean titers (error bars indicate the 95% CIs) of the anti-RBD IgG against the SARS-CoV-2 spike protein. The dashed line indicates the 506 BAU/mL cutoff value. (**b**) Geometric mean (error bars indicate the 95% CIs) of percent neutralizing antibody inhibition as measured by surrogate viral neutralization test. (**c**) Geometric mean (error bars indicate the 95% CIs) of IFN-γ-producing T-cell response to the S1 scanning peptide pool. The dashed line indicates the 6 SFU/10^6^ PBMCs cutoff value. _A_b, antibody; BAU, binding antibody units; CI, confidence interval; HD, hemodialyzed patients; IgG, immunoglobulin G; PBMCs, peripheral blood mononuclear cells; PD, peritoneal dialyzed patients; RBD, receptor-binding domain; S1, S1 domain of the spike protein; SFU, spot-forming unit; SARS-CoV-2, severe acute respiratory syndrome coronavirus-2.

**Figure 3 vaccines-10-01064-f003:**
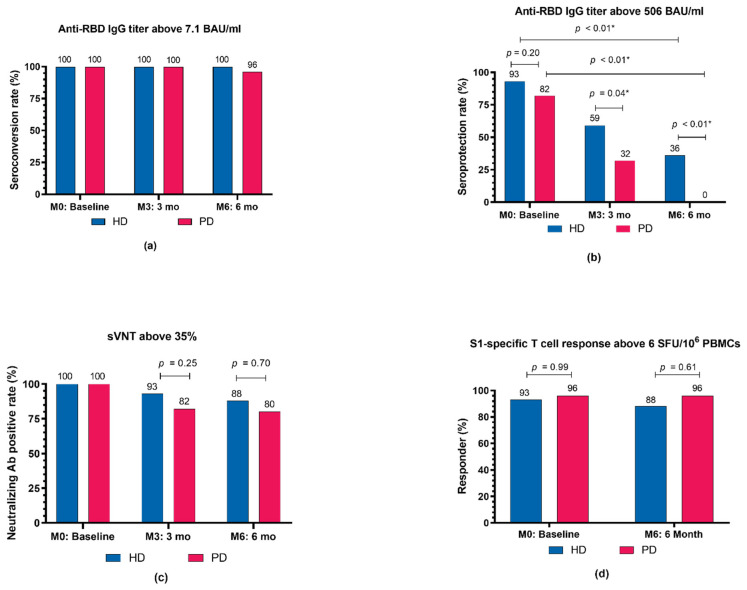
Proportions of dialysis patients who achieved specific immunogenicity thresholds following COVID-19 vaccination with an extended primary series, composed of a heterologous inactivated SARS-CoV-2 prime-boost and ChAdOx1 nCoV-19 vaccine as the third dose, at 2 weeks (M0), 3 months (M3), and 6 months (M6) after the completion of vaccination. (**a**) Proportions of patients who met the seroconversion threshold (anti-RBD IgG titer of ≥7.1 BAU/mL). (**b**) Proportions of patients who met the seroprotection threshold against the SARS-CoV-2 Alpha (B.1.1.7) variant (anti-RBD IgG titer of ≥506 BAU/mL). (**c**) Proportions of patients who met the threshold of neutralizing antibody positive (percent inhibition of >35%). (**d**) Proportions of patients who met the criteria for a S1-specific T-cell responder (IFN-γ-producing T-cell response to the S1 scanning peptide pool of >6 SFU/10^6^ PBMCs). Ab, antibody; BAU, binding antibody units; HD, hemodialyzed patients; IgG, immunoglobulin G; PBMCs, peripheral blood mononuclear cells; PD, peritoneal dialyzed patients; RBD, receptor-binding domain; S1, S1 domain of the spike protein; SFU, spot-forming unit; sVNT, surrogate virus neutralization test; * *p* < 0.05.

**Table 1 vaccines-10-01064-t001:** Baseline demographic data and laboratory parameters of study participants.

Demographic Data n (%) or Mean (SD)	HD (n = 29)	PD (n = 28)	Controls (n = 14)	*p*-Value
Age [years], mean (SD)	44.3 (9.7)	41.0 (11.5)	43.6 (8.0)	0.46
Male sex, n (%)	22 (76%) *	17 (61%)	5 (36%)	0.04
Body mass index [kg/m^2^], mean (SD)	25.4 (5.2)	23.6 (4.4)	26.5 (6.1)	0.20
Age-adjusted Charlson Comorbidity Index, median (IQR)	3 (3–5)	2.5 (2–4)	N/A	0.06
Comorbidities, n (%)				
Diabetes mellitus	13 (45%) *	7 (25%)	1 (8%)	0.05 ^a^
Hypertension	22 (76%) *	25 (89%) *	2 (17%)	<0.01 ^a^
Dyslipidemia	11 (38%)	9 (32%)	4 (33%)	0.94 ^a^
Ischemic heart disease	7 (24%)	2 (7%)	0 (0%)	0.07
Etiologies of ESKD, n (%)				0.15
Diabetes	6 (21%)	5 (18%)	N/A	
Hypertension	3 (10%)	8 (29%)	N/A	
Chronic glomerulonephritis	5 (17%)	7 (25%)	N/A	
Other	3 (10%)	0 (0%)	N/A	
Unknown	12 (41%)	7 (25%)	N/A	
Immunosuppressive drugs	0 (0%)	1 (3%)	0 (0)	
Dialysis vintage, months, median (IQR)	32.6 (19.0–83.5)	34.1 (7.2–57.2)	N/A	0.18
Total Kt/V_urea_	1.6 (0.3%)	2.0 (0.4%)	N/A	N/A
Anuria, n (%)	16 (55%)	9 (32%)	N/A	0.04
Smoking, n (%)	13 (45%)	10 (36%)	0 (0)	0.60
Baseline lab results				
White blood cell count [×10^9^ cells/L], mean (SD)	7.0 (1.9)	7.3 (2.8)	7.8 (2.7) ^b^	0.69
Absolute lymphocyte count [×10^9^ cells/L], mean (SD)	1.6 (0.5) *	1.5 (0.8) *	2.3 (0.9) ^b^	0.03
Hemoglobin [g/dL], mean (SD)	11.1 (2.1) *	10.0 (2.3) *	13.5 (1.1) ^b^	<0.01
Calcium [mg/dL], mean (SD)	8.8 (1.0)	8.6 (1.0)	N/A	0.45
Phosphorus [mg/dL], mean (SD)	5.5 (1.9)	5.4 (1.9)	N/A	0.87
Albumin [g/L], mean (SD)	40.1 (4.3)	33.1 (4.1)	N/A	<0.001
Ferritin [ng/mL], median (IQR)	304 (119–441)	367 (156–751)	N/A	0.04
Time interval between vaccination and immunogenicity evaluation				
Time interval from M0 to M3 [months], mean (SD)	3.5 (0.2) *	3.4 (0.2) *	4.2 (0.3)	<0.01
Time interval from M0 to M6 [months], mean (SD)	6.6 (0.3) ^c^	6.4 (0.3) ^c^	N/A	0.28

A Student’s *t*-test and Fisher’s exact test were used for comparing continuous and categorical variables, respectively, between patients undergoing HD and PD. BMI was computed by dividing the participant body weight in kilograms by the square of their height in meters. Total Kt/V represents total small-solute clearances. ESKD, end-stage kidney disease; HD, hemodialyzed patients; IQR, interquartile range; n, number; N/A, not applicable; PD, peritoneal dialyzed patients; SD, standard deviation. ^a^, Fisher’s exact test; ^b^, evaluated in 7 participants; ^c^, evaluated in 25 participants. * *p* < 0.05 (compared with controls).

**Table 2 vaccines-10-01064-t002:** Baseline characteristics and predictors of anti-RBD IgG seroprotection loss (anti-RBD IgG titer below 506 BAU/mL) at 3 months (M3) after an additional dose of the ChAdOx1 nCoV-19 vaccine among 57 patients undergoing dialysis.

Variable n (%) or Mean (SD)	Anti-RBD IgG Titer ≥ 506 BAU/mL at M3 (Seroprotection) (n = 26)	Anti-RBD IgG Titer < 506 BAU/mL at M3 (Non-Sero Protection) (n = 31)	Univariate Analysis		Multivariate Analysis	
			OR (95%CI)	*p*-Value	OR (95%CI)	*p*-Value
Age [years], mean (SD)	39.6 (9.2)	45.3 (11.3)	1.05 (1.00–1.11)	0.05	1.10 (1.02–1.18)	0.01
Peritoneal dialysis, n (%)	9 (35%)	19 (61%)	2.99 (1.01–8.84)	0.05	5.23 (0.87–31.50)	0.07
Dialysis duration [months], median (IQR)	45.0 (25.0–85.1)	32.0 (7.46–53.3)	0.98 (0.98–1.00)	0.06	0.99 (0.98–1.00)	0.15
Anti-RBD IgG titer at M0 [BAU/mL], mean (SD) Tertile 1 (ref.) Tertile 2 Tertile 3	2216 (1540–4172) 1045 (89) 1746 (328) 4336 (820)	1010 (510–1364) 612 (320) 1490 (419) 2658 (171)	1.0 0.17 (0.04–0.70) 0.07 (0.15–0.34)	0.01 <0.01	1.0 0.15 (0.03–0.77) 0.06 (0.01–0.38)	0.02 <0.01
Number(s) of patients with an absolute lymphocyte count of ≥1.5 × 10^9^ cells/L, n (%)	15 (58%)	15 (48%)	0.69 (0.24–1.96)	0.48		
Albumin [g/L], mean (SD)	37.7 (5.6)	35.9 (5.4)	0.94 (0.85–1.04)	0.23	1.10 (0.94–1.30)	0.23
Number(s) of patients with serum ferritin of ≥500 ng/mL, n (%)	5 (19%)	10 (32%)	2.0 (0.58–6.86)	0.27		

The odds ratios (ORs) indicate the association between each variable and loss of anti-RBD IgG seroprotection status at 3 months after vaccination completion (anti-RBD IgG titer of <506 BAU/mL). BAU, binding antibody units; CI, confidence interval; IgG, immunoglobulin G; IQR, interquartile range; M0, 2 weeks after an additional dose of the ChAdOx1 nCoV-19 vaccine; M3, 3 months after an additional dose of the ChAdOx1 nCoV-19 vaccine; N, number; OR, odds ratio; RBD, receptor-binding domain; SD, standard deviation.

## Data Availability

Owing to privacy and ethical concerns, the datasets collected and/or analyzed during this work are not publicly available; however, anonymized data are available upon reasonable request from the corresponding author.

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
