# Peer review of "Durability of Humoral and Cellular Immunity after an Extended Primary Series with Heterologous Inactivated SARS-CoV-2 Prime-Boost and ChAdOx1 nCoV-19 in Dialysis Patients (ICON3)"

_vaccines, 2022, doi:10.3390/vaccines10071064_

Round 1

Reviewer 1 Report

This is a very nice job, though there are some weaknesses and limitations in this paper, such as small patient size.  This kind of studies should be encouraged and continually supported. I have only some minor comments related to accuracy, concise and clear expression.  Some of the questions are pointed out here, but it would be better the authors check the entire manuscript, so the reading could be smoother.

1. The term of "inactivated vaccine" is used in many places in the manuscript. This is not accurate scientifically, though can be seen in some news. This term should be avoided in the paper. It is not the vaccine inactivated (inactivated vaccine cannot be used for immunization), it is the inactivated SARS-CoV-2 virus, or CoronaVac vaccine.

2. In line 82-84: "Patients who were … received any immunizations (COVID-19 or others) during the study were excluded from the analysis." This is hard to understand as all patients were received vaccines in the study.

3. Line 186-188: "The median anti-RBD IgG titer at M3 in the HD group was significantly 186 higher compared with the PD group [632 (337–1526) BAU/ml vs. 368 (217–5540) BAU/ml, 187 p = 0.02](,) but not compared with the control group [482 (196–658) BAU/ml, p = 0.09]."

This sentence is not clear. The authors may want to tell us that the median anti-RBD IgG titer at M3 of the control group is 482 (196–658) BAU/ml, which is not significantly different from both the HD and PD groups. (?)

4. Line 189-192: "The mean (SD) percent reduction of anti-RBD IgG from M0 to M3 in patients undergoing HD was similar to that in patients undergoing PD [−59.5% (27.6%) vs. −69.1% (17.1%), p = 0.13](,) but lower than that in control participants [−80.2% (11%), p = 0.01]"

This sentence can be also improved. Is only the number of HD or both HD and PD lower than that of the control group? Should -80.2% is lower than -59.5% and -69.1%?

5. Moving the subsection title "3.3 SARS-CoV-2-specific CMI" to the left end in line 271.

6. Line 312-313: "PD patients exhibited higher vaccine-induced antibody levels than did HD patients in several trials, but not all [31, 32]." (?)

7. Line 333: Immune protection against COVID-19 is inversely proportional to the neutralizing antibody levels [35], As with the anti-RBD seroconversion rate, the neutralizing antibody positivity rate declined over time.

Author Response

Reviewer 1

This is a very nice job, though there are some weaknesses and limitations in this paper, such as small patient size.  This kind of studies should be encouraged and continually supported. I have only some minor comments related to accuracy, concise and clear expression.  Some of the questions are pointed out here, but it would be better the authors check the entire manuscript, so the reading could be smoother.

Answer: Thank you very much. We appreciate your comments. We have revised a few sentences to improve the readability of the manuscript.

  1. The term of "inactivated vaccine" is used in many places in the manuscript. This is not accurate scientifically, though can be seen in some news. This term should be avoided in the paper. It is not the vaccine inactivated (inactivated vaccine cannot be used for immunization), it is the inactivated SARS-CoV-2 virus, or CoronaVac vaccine.

Answer: We have changed the "inactivated vaccine" throughout the entire manuscript to "inactivated SARS-CoV-2 vaccine."

  1. In line 82-84: "Patients who were … received any immunizations (COVID-19 or others) during the study were excluded from the analysis." This is hard to understand as all patients were received vaccines in the study.

Answer: The sentence above has been edited for clarity. Please see page 2.

Line 83-85: “Patients who received COVID-19 immunizations outside of the study or were diagnosed with COVID-19 infection during the study were excluded from the analysis.”

  1. Line 186-188: "The median anti-RBD IgG titer at M3 in the HD group was significantly 186 higher compared with the PD group [632 (337–1526) BAU/ml vs. 368 (217–5540) BAU/ml, 187 p = 0.02](,)but not compared with the control group [482 (196–658) BAU/ml, p = 0.09]."

This sentence is not clear. The authors may want to tell us that the median anti-RBD IgG titer at M3 of the control group is 482 (196–658) BAU/ml, which is not significantly different from both the HD and PD groups. (?)

Answer: Thank you for your comment. We have revised the above sentences as suggested. Please see page 6.

Line 191-194: “The median (IQR) anti-RBD IgG titer at M3 in the HD group was significantly higher compared with the PD group [632 (337–1526) BAU/ml vs. 368 (217–5540) BAU/ml, p = 0.02]. In the control group, the median (IQR) anti-RBD IgG titer at M3 was 482 (196–658) BAU/ml, which did not significantly differ from either the HD or PD groups.”

  1. Line 189-192: "The mean (SD) percent reduction of anti-RBD IgG from M0 to M3 in patients undergoing HD was similar to that in patients undergoing PD [−59.5% (27.6%) vs. −69.1% (17.1%), p = 0.13](,) but lower than that in control participants [−80.2% (11%), p = 0.01]"

This sentence can be also improved. Is only the number of HD or both HD and PD lower than that of the control group? Should -80.2% is lower than -59.5% and -69.1%?

Answer: The above sentences have been rewritten as suggested. Please see page 6.

Line 195-201: “The mean (SD) percent reduction in anti-RBD IgG titer from M0 to M3 was greatest in the control group at ­−80.2% (11%). Compared to the control group, the mean (SD) percent reduction in anti-RBD IgG titer from M0 to M3 was significantly lower in the HD group [−59.5% (27.6%), p = 0.01)], whereas it was marginally lower in the PD group [−69.1% (17.1%), p = 0.07]. The mean (SD) percent reduction of anti-RBD IgG from M0 to M3 in patients undergoing HD was comparable to that in patients undergoing PD (p = 0.13) (Supplementary Table S1).”

  1. Moving the subsection title "3.3 SARS-CoV-2-specific CMI" to the left end in line 271.

Answer: Thank you. We have moved the subsection title "3.3 SARS-CoV-2-specific CMI" to a proper position. Please see line 287.

  1. Line 312-313: "PD patients exhibited higher vaccine-induced antibody levels than did HD patients in several trials, but not all [31, 32]." (?)

Answer: We have revised the above sentences as suggested. Please see pages 10-11.

Line 330-332: “The effect of dialysis modality on SARS-CoV-2-vaccine-induced antibody levels remains uncertain. In several studies, PD patients exhibited higher vaccine-induced antibody levels than HD patients [31, 32].”

  1. Line 333: Immune protection against COVID-19 is inversely proportional to the neutralizing antibody levels [35], As with theanti-RBD seroconversion rate, the neutralizing antibody positivity rate declined over time.

Answer: We have corrected the grammatical error as suggested. Please see page 11.

Line 356: “…….to the neutralizing antibody levels [35]. As with the anti-RBD seroconversion rate….”

Reviewer 2 Report

I think that the number of subjects is too small to conclude with 29 HD, 28 PD, and 14 controls.

The manuscript describes the durability of humoral and cellular immunity after two-dose CoronaVac vaccine followed by one dose of AZD1222 in Dialysis patients and humoral immunity declined substantially in dialysis patients, while cell-mediated immunity remained stable 6 months after the extended heterologous primary series of two inactivated SARS-CoV-2/ChAdOx1 nCoV-19 vaccine.

Major

1. The study is well designed and described, but the numbers of participants in each group are too small to compare with each other and draw conclusions. (29 HD, 28 PD, 14 controls)

2. Sampling timing after vaccination is the most important issue in monitoring antibody titers. Please describe the time point of blood collection after booster vaccination (M0, M3, M6) as median and range/quartile in each group (HD, PD, control) in Table 1. If the median sampling time of HD group in M3 is 3 months and the median sampling time of PD group is 3.5 months, the median antibody titer of HD group can be higher than that of PD group.

Author Response

Reviewer 2

I think that the number of subjects is too small to conclude with 29 HD, 28 PD, and 14 controls.

The manuscript describes the durability of humoral and cellular immunity after two-dose CoronaVac vaccine followed by one dose of AZD1222 in Dialysis patients and humoral immunity declined substantially in dialysis patients, while cell-mediated immunity remained stable 6 months after the extended heterologous primary series of two inactivated SARS-CoV-2/ChAdOx1 nCoV-19 vaccine.

Major

  1. The study is well designed and described, but the numbers of participants in each group are too small to compare with each other and draw conclusions. (29 HD, 28 PD, 14 controls)

 Answer: We have acknowledged our study limitation on page 12.

Line 387-389: “Our study has several limitations. Due to the small number of patients, its findings and conclusions should be considered preliminary. Further studies with larger sample sizes are required to determine definitive conclusions.”

In addition, we have revised our conclusions as follows:

Please see page 12, lines 406-408: “Further studies with larger populations are encouraged to better generalize our results and evaluate the clinical effectiveness of this vaccine regimen.”

  1. Sampling timing after vaccination is the most important issue in monitoring antibody titers. Please describe the time point of blood collection after booster vaccination (M0, M3, M6) as median and range/quartile in each group (HD, PD, control) in Table 1. If the median sampling time of HD group in M3 is 3 months and the median sampling time of PD group is 3.5 months, the median antibody titer of HD group can be higher than that of PD group.

Answer: We agree with your insightful comment. As suggested, we have added the information regarding the time interval between vaccination and immunogenicity evaluation to Table 1 on page 5.

Round 2

Reviewer 2 Report

none